# Purines and Adenosine Receptors in Osteoarthritis

**DOI:** 10.3390/biom13121760

**Published:** 2023-12-07

**Authors:** Bruce N. Cronstein, Siddhesh R. Angle

**Affiliations:** 1Divisions of Rheumatology and Precision Medicine, New York University Grossman School of Medicine, New York, NY 10016, USA; 2Regenosine, Inc., Princeton, NJ 08540, USA; sid@regenosine.com

**Keywords:** Osteoarthritis, adenosine receptors, purine metabolism

## Abstract

OA is a common and debilitating condition that restricts mobility and diminishes the quality of life. Recent work indicates that the generation of adenosine at the cell surface is an important mediator of chondrocyte homeostasis, and topical application of adenosine in a slow-release form (liposomes) can halt the progression of OA and diminish the pain associated with OA. Here, we review the evidence indicating that adenosine, acting at A_2A_ receptors, plays a critical role in endogenous and exogenous treatment and reversal of OA.

## 1. Osteoarthritis

Osteoarthritis (OA), a form of degenerative arthritis, is the most common type of arthritis. OA affects as many as 54 million people in the United States and 528 million people worldwide (http://ghdx.healthdata.org/gbd-results-tool accessed on 15 October 2023). The main symptoms of OA include pain, stiffness, and swelling of the affected joint, leading to increasing disability. Age, female gender, prior trauma to the affected joint, anatomic factors, and genetics may all contribute to the development of OA. Inflammation, osteophytes, bone changes, and loss of cartilage are all present in the joints of people with OA [1].

A leading cause of disability worldwide, the disease drives millions to seek therapeutic interventions. However, at present, there is no definitive medical treatment to prevent or reverse the progression of OA. The injection of corticosteroids and hyaluronic acid provides temporary relief without altering the course of the disease, and the number of joint replacements has increased dramatically and is expected to continue to increase [1].

The pathogenesis and treatment of OA have been studied in a variety of rodent models. As in humans, trauma may lead to the development of OA, and there are a number of post-traumatic OA models in which the affected joint is surgically disrupted. Age and degenerative models of OA are also used to study the treatment and pathogenesis of OA (obesity and intra-articular injection of monosodium iodoacetate) [2]. Various gene knockouts also develop spontaneous OA. All of these models have their flaws and advantages, and since no new therapies have been introduced based on the use of these models, it is difficult to determine the most useful model of OA. As discussed below, adenosine and adenosine receptor agonists have been studied in mice, rats, and dogs using spontaneous gene knockout), post-traumatic, and degenerative OA models [3,4].

## 2. Purine Metabolism, Receptors, and OA

The central player in OA is the chondrocyte. Chondrocytes respond to excess mechanical loading by releasing inflammatory mediators and proteolytic enzymes causing further cartilage damage [5]. In 2002, Tesch and colleagues reported that equine chondrocytes respond to adenosine A2 receptor stimulation and, in subsequent experiments, reported that the treatment of cartilage explants with adenosine deaminase led to a spontaneous expression of proteases associated with cartilage degeneration and the development of OA [6,7]. The interpretation of these results is complicated since cellular levels of adenosine deaminase in equine tissues approached those reported for children with adenosine deaminase deficiency and adenosine deaminase activity is undetectable in equine plasma [8,9,10,11]. The minimal adenosine deaminase levels in equine chondrocytes and plasma suggests that chondrocytes and cartilage might be disproportionately affected by endogenous adenosine levels.

In subsequent studies, it was reported that patients with a hereditary absence of CD73 activity with, presumably, lower adenosine levels at the cell surface lead to the premature development of osteoarthritis [12,13]. Mice lacking CD73 activity also develop spontaneous OA [4]. Similarly, mice lacking A_2A_ receptors [4] and A_3_ receptors [14] develop spontaneous OA. Moreover, the intra-articular administration of a selective A_2A_ agonist [3,4] and the administration of a selective A3 receptor agonist prevent the development of OA in murine models [14,15,16]. The anti-inflammatory effects of A3 receptors likely mediate the effects of the agent used on the development of OA [4]. Similarly, the deletion of A_2A_ receptors leads, in mice, to the spontaneous development of OA [4], as does the deletion of CD73, in both mice and humans [4], but the effects of loss of A_2A_ receptors also involve changes in chondrocyte physiology [3,4,17,18]. Moreover, with age and inflammation, chondrocytes have a reduced capacity to synthesize and maintain adenosine triphosphate (ATP). The levels of ATP and adenosine, its metabolite, fall after treatment of mouse chondrocytes and rat tibia explants with interleukin-1β (IL-1β), an inflammatory mediator that contributes to OA [4]. However, restoring the levels of adenosine with the intra-articular (IA) injection of long-acting adenosine with liposomal adenosine or adenosine-functionalized polymeric nanoparticles prevents the development of OA in a rat model of post-traumatic OA [19] and reverses cartilage damage in both a rat model of post-traumatic OA and an obesity-induced model of OA in mice via interaction with A_2A_ receptors [3,4]. Furthermore, the IA administration of liposomal adenosine in dogs with OA, after medial meniscus release, led to a steady decrease in pain, with a corresponding gain in function for up to 6mo after treatment. Moreover, injections improved comfortable range of motion (CROM) to near-normal values at 6 mo. Compared to the placebo, the treatment also improved radiographic markers, and MRIs showed a significant slowing in OA progression, including improvement in cartilage morphology, synovitis, and osteophyte occurrence [20].

## 3. The Mechanisms by Which Adenosine Mitigates OA

Extracellular adenosine derives mainly from the hydrolysis of ATP (primarily, but not exclusively, by the ectoenzymes CD39 and CD73) and mediates its effects via the activation of G-protein-coupled receptors (A_1_, A_2A_, A_2B_ and A_3_). However, the chondroprotective effects of long-acting adenosine are exclusively mediated via A_2A_ receptors [4]. A_2A_ stimulation mediates its effects on chondrocytes, cartilage, and OA via multiple mechanisms. The deletion of A_2A_ leads to the differential expression of genes associated with senescence, aging, and inflammation in neonatal murine chondrocytes [21]. In addition, A_2A_ stimulation resensitizes TGF-β receptors, which, in OA, do not transmit signals in chondrocytes [3], promotes mitophagy, increases mitochondrial function (and ATP production [18]), and stimulates autophagy via a FoxO-dependent mechanism [22]. When studied in vitro, A_2A_ stimulation prevents both spontaneous and oxidant-mediated cellular senescence in chondrocytic cells, diminishes the nuclear localization of the senescence markers/mediators p16 and p21, reduces the cellular expression of full-length p53, and increases the expression of the D133p53 splice variant of p53, the expression of which has potent anti-senescent effects [23]. Moreover, as noted above, in chondrocytes from A_2A_-deficient mice, there is an increased expression of genes associated with cellular senescence, downregulation of autophagy, enhanced expression of inflammatory genes, increased expression of genes coding for proteases that digest the matrix, and diminished matrix protein gene expression [21]. These findings are consistent with the hypothesis that autocrine activation of A2A is critical for the maintenance of chondrocyte and cartilage homeostasis (Figure 1).

In contrast to the reported beneficial effects of adenosine on cartilage and chondrocytes, Mistry and colleagues reported that the treatment of susceptible mice with an adenosine deaminase inhibitor (thereby inducing an increase in adenosine levels) led to injury and death to chondrocytes with development of OA and in vitro studies demonstrated that this effect required adenosine uptake [24]. In studies of the mechanism of action by which shockwave therapy ameliorates OA, Tan and colleagues found that the A_2B_ receptor expression is increased and that A_2B_ receptor stimulation reduces chondrocyte differentiation in human mesenchymal stem cells [25]). These studies further suggested that the A_2B_ receptor-mediated effects on chondrogenesis were linked to IL-6 expression and signaling. It should be noted that children lacking adenosine deaminase activity (with a marked increase in extracellular levels) do not develop OA but have short growth plates with few proliferating chondrocytes and chondrocyte necrosis [26]. Thus, extreme adenosine levels, such as those encountered in adenosine deaminase-deficient children, in the setting of chondrocyte injury or extracorporeal shockwave therapy may lead to diminished chondrogenesis via the stimulation of A_2B_ receptors.

Further confirmation of the OA-protective role of adenosine and its receptors in cartilage is provided in a recent observational study in which patients treated with ticagrelor, a P2_Y12_ antagonist that also blocks adenosine uptake and increases local adenosine levels, were less likely to develop OA than patients treated with clopidogrel, a P_2Y12_ antagonist that does not inhibit adenosine uptake [27], a finding consistent with the role of adenosine in maintaining chondrocyte homeostasis. Other studies have suggested that increased ingestion of caffeine, a common constituent of food that is a non-selective adenosine receptor antagonist, is a risk factor for the development of OA (recently reviewed in [28]). Istradefylline is a selective A_2A_ receptor antagonist used to treat Parkinson’s Disease and, based on the discussion above, would be expected to promote the development of OA. Insufficient data regarding the long-term effects of this drug, e.g., OA, have not been fully established.

## 4. Conclusions

The findings reviewed here strongly support the hypothesis that adenosine, acting at A2A and A_3_ receptors, plays a central role in maintaining chondrocyte homeostasis. Moreover, these studies suggest that adenosine receptors may be a new therapeutic target in the treatment of OA.

## Figures and Tables

**Figure 1 biomolecules-13-01760-f001:**
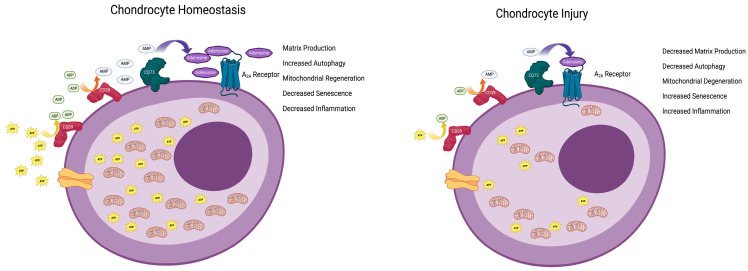
Endogenous adenosine maintains chondrocyte homeostasis. ATP is transported into the extracellular space where it is sequentially dephosphorylated via the cell surface enzymes CD39 and CD73 (Ectonucleoside triphosphate diphosphohydrolase 1 and 5-nucleotidase, respectively) to adenosine. Adenosine stimulates the A_2A_ adenosine receptor to diminish cellular aging and apoptosis and increase autophagy, mitophagy, and inflammation in chondrocytes, thereby promoting homeostasis. Following injury, there is less ATP transported into the extracellular space, leading to diminished adenosine concentrations resulting in diminished autophagy, mitophagy, and inflammation with increased senescence and apoptosis. Figure created with BioRender.com.

## Data Availability

Not applicable.

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
