# Peer review of "Purines and Adenosine Receptors in Osteoarthritis"

_biomolecules, 2023, doi:10.3390/biom13121760_

Round 1

Reviewer 1 Report

Comments and Suggestions for Authors

In this Opinion manuscript: the evidence indicating that adenosine, acting at A2A receptors, plays a critical role in endogenous and exogenous treatment and reversal of osteoarthritis.

The manuscript is good. I have a few minor comments. 

Overall, the write up needs English editing.

Few places needs references. For example, the OA paragraph.

Authors should clarify the in vivo studies weather that used degenerative OA animal model or Traumatic OA models.

A few lines about Post-traumatic OA in the introduction will be great. 

Comments on the Quality of English Language

Few sentences are long and without proper breaks . For example, Line 23/24, Line 86-90

Author Response

We thank the reviewer for the thoughtful critiques and believe that the response to these critiques has improved the manuscript. 

Overall, the write up needs English editing. We have edited the language throughout the manuscript.

Few places needs references. For example, the OA paragraph. This subject has been well reviewed over the years and we have included a reference to a general review of OA. In addition we have included a review of animal models of OA.  

Authors should clarify the in vivo studies weather that used degenerative OA animal model or Traumatic OA models. We have now noted in the third paragraph of the Introduction that adenosine has been tested as a therapy for OA in degenerative and post-traumatic arthritis.  

A few lines about Post-traumatic OA in the introduction will be great. We have added a brief discussion of post-traumatic OA in the Introduction. 

Reviewer 2 Report

Comments and Suggestions for Authors

This short review manuscript is well structured and written, as would be expected from a (the) leading expert in the area. I would challenge the authors with the following comments/suggestions:

-the data in equine models (protection afforded by A3 receptor activation) seem at odds with the murine data (protection afforded by A2A receptor activation); it would be of interest to briefly comment which animal model models best the human condition.

-given the prevalence of the consumption of caffeinated coffee, and given that caffeine is an adenosine receptor antagonist, is there any evidence for a positive association between coffee intake and OA? I would expect that this should be a mandatory aspect to be briefly presented and commented. 

-do OA patient with cardiac comorbidities, and treated with dipyridamole, present a better OA prognosis?

-are there any preliminary data, in PD patients treated with istradephyllne, of a precipitation of OA?

-given the proposed key role of A2AR, are there associations between A2AR polymorphisms and the incidence and/or evolution of OA?

Author Response

We thank the reviewer for his/her insights and comments. Following are the responses to the reviewers' comments:

-the data in equine models (protection afforded by A3 receptor activation) seem at odds with the murine data (protection afforded by A2A receptor activation); it would be of interest to briefly comment which animal model models best the human condition. It is difficult to determine which of the animal models most resembles the human condition as, to date, there have been no drugs translated from animals to humans. Nonetheless, in the 4 species tested (mouse, rat, dog and horse) the effects of adenosine on osteoarthritis in two different types of osteoarthritis have paralleled each other.  The A3 receptor findings were made in a murine model in which monosodium iodoacetate is injected intra-articularly. This model is one in which inflammation is the dominant finding.

-given the prevalence of the consumption of caffeinated coffee, and given that caffeine is an adenosine receptor antagonist, is there any evidence for a positive association between coffee intake and OA? I would expect that this should be a mandatory aspect to be briefly presented and commented. 

The reviewer is correct and there is actually a large literature on the effects of caffeine on osteoarthritis and chondrocytes. This has recently been reviewed in [28] although the effects of caffeine mediated by adenosine receptor antagonism is not well covered in this review.  We have commented on this on page 5. 

-do OA patient with cardiac comorbidities, and treated with dipyridamole, present a better OA prognosis? Dipyridamole was tested (and failed) as a treatment for osteoarthritis although we were unable to find any suggestion that patients taking dipyridamole for cardiac disease we did find evidence for a different ent1 antagonist on the development of OA. A recent paper describes an observational study of ticagrelor, an anti-platelet drug that also blocks ent1, on the development of OA in patients. The authors found that ticagrelor but not clopidogrel (another P2Y12 antagonist without effects on adenosine uptake inhibited the development of OA, This finding is consistent with the notion that the reviewer's undelying hypothesis, increasing extracellular adenosine levels with either dipyridamole or, in this case ticagrelor, blocks the development of OA.

-are there any preliminary data, in PD patients treated with istradephyllne, of a precipitation of OA? This is an important question but we were unable to find any evidence one way or the other.

-given the proposed key role of A2AR, are there associations between A2AR polymorphisms and the incidence and/or evolution of OA? We were unable to find any association of A2AR polymorphisms and development of OA in the literature. 

Reviewer 3 Report

Comments and Suggestions for Authors

The current Opinion from Bruce N. Cronstein and Siddhesh R. Angle is an interesting document resuming some of the major findings concerning the relevance of the adenosinergic system in cartilage homeostasis and osteoarthritis (OA).

My suggestions to improve this document:

1)      P1 receptors should be as follows (according to IUPHAR nomenclature): A1, A2A, A2B, A3 (please correct this in both text and figure); the same for P2Y12 receptor;

2)      Include a brief comment on both A1 and A2B receptors; several works suggest a relevant role in cartilage homeostasis and in OA (e.g., https://doi.org/10.1111/joa.12530; https://doi.org/10.1007/s00424-014-1529-8; https://doi.org/10.1038/s41598-017-14875-y). In particular, the A2B receptor seems to play a role on cartilage remodelling providing that high enough levels of adenosine are generated in the extracellular microenvironment as a consequence of stressful conditions, which include trauma, inflammation and hypoxia. In fact, in vitro and in vivo mice models show that shockwave treatment promotes HIF-1α gene expression, and that adenosine A2B receptor overexpression may be linked to HIF-1α levels in osteoarthritic joints. Please include a brief comment on these findings.

Author Response

1)      P1 receptors should be as follows (according to IUPHAR nomenclature): A1, A2A, A2B, A3 (please correct this in both text and figure); the same for P2Y12 receptor;

We thank the reviewer for pointing this out and have made the appropriate changes in the text and the figure.

2)      Include a brief comment on both A1 and A2B receptors; several works suggest a relevant role in cartilage homeostasis and in OA (e.g., https://doi.org/10.1111/joa.12530; https://doi.org/10.1007/s00424-014-1529-8; https://doi.org/10.1038/s41598-017-14875-y). In particular, the A2B receptor seems to play a role on cartilage remodelling providing that high enough levels of adenosine are generated in the extracellular microenvironment as a consequence of stressful conditions, which include trauma, inflammation and hypoxia. In fact, in vitro and in vivo mice models show that shockwave treatment promotes HIF-1α gene expression, and that adenosine A2B receptor overexpression may be linked to HIF-1α levels in osteoarthritic joints. Please include a brief comment on these findings.

We have now included a discussion of the role of A2B receptors in the pathogenesis of OA.  Although A2B receptor overexpression and stimulation may play a role in the amelioration of OA it would seem that the evidence would also be consistent with the role of A2B receptors as mediators of chondrocyte injury and the development of OA. A1 receptors have been linked to pain relief in many settings but we did not add that here. The role of A1 receptors in the development is not at all clear from the data reported by Choi, et al. It is difficult for this reviewer to understand how addition of a selective A1 receptor antagonist can abrogate the effects of a pan-adenosine receptor antagonist and suggests that there are other metabolic effects of either DPCPX or caffeine or both that explain the abrogation of the caffeine effects by DPCPX.